# TEST-TIME TRAINING FOR OUT-OF-DISTRIBUTION INDUSTRIAL ANOMALY DETECTION VIA ROBUST DISTRIBUTION ALIGNMENT

## ABSTRACT

Detecting anomalous patterns is essential for quality control in industrial applications, with state-of-the-art methods relying on large defect-free datasets to model normal distributions. However, robustness under domain shift, such as changes in lighting or sensor drift, remains a critical challenge in real-world deployment. An existing work, Generalized Normality Learning (GNL), addresses domain shifts by enforcing feature consistency through training-time augmentation, but its reliance on prior knowledge of target distributions and access to training data at inference limits flexibility. To overcome these limitations, we propose a memory bank-based anomaly detection method that avoids retraining or access to training data during inference. We improve the robustness to distribution shifts via distribution alignment based test-time training. Our approach leverages a modified Sinkhorn distance to align distributions and handle outliers, offering a more resilient solution for industrial anomaly detection under realistic constraints. Extensive evaluations on out-of-distribution anomaly detection benchmarks demonstrate the effectiveness.

## 1 INTRODUCTION

Detecting anomalous patterns is critical for ensuring quality control in industrial applications. State-of-the-art methods for industrial anomaly detection often rely on large defect-free training samples to model the distribution of normal patterns using techniques such as generative models Deng & Li (2022); Zhang et al. (2023b) or memory banks Roth et al. (2022); Xie et al. (2023); Gu et al. (2023); Hu et al. (2024). These approaches have achieved remarkable performance on various industrial anomaly detection datasets, giving the impression that the problem is largely solved. However, one key issue that remains overlooked is the robustness of these methods, which is vital for real-world deployment. Among the many challenges to robustness, domain shift—a mismatch between the data distributions of training and testing sets—is particularly common in industrial settings, arising from factors like changes in lighting or sensor drift.

A pioneering work, generalized normality learning (GNL)Cao et al. (2023), tackled this issue by treating anomaly detection under distribution shift as an out-of-distribution (OOD) generalization problem. GNL aims to improve model generalization to testing data that deviates from the training distribution. During training, GNL encourages consistency in the intermediate features of augmented normal samples, ensuring that the model's representation is less sensitive to shifts in the data distribution at test time. For inference, GNL utilizes exact feature distribution matching (EFDM)Zhang et al. (2022) to align testing samples with randomly sampled normal data from the training set, achieving superior results on corrupted test datasets.

Despite these advancements, we identify two key limitations in the current approach. First, requiring prior knowledge of the target data distribution during training may not be practical. GNL's performance can degrade when the distribution shift at test time differs significantly from the augmentations used during training. Second, accessing normal training samples at inference may be restricted due to privacy concerns or data storage constraints. Thus, a more flexible approach to industrial anomaly detection is needed. We propose two critical constraints for an effective solu-

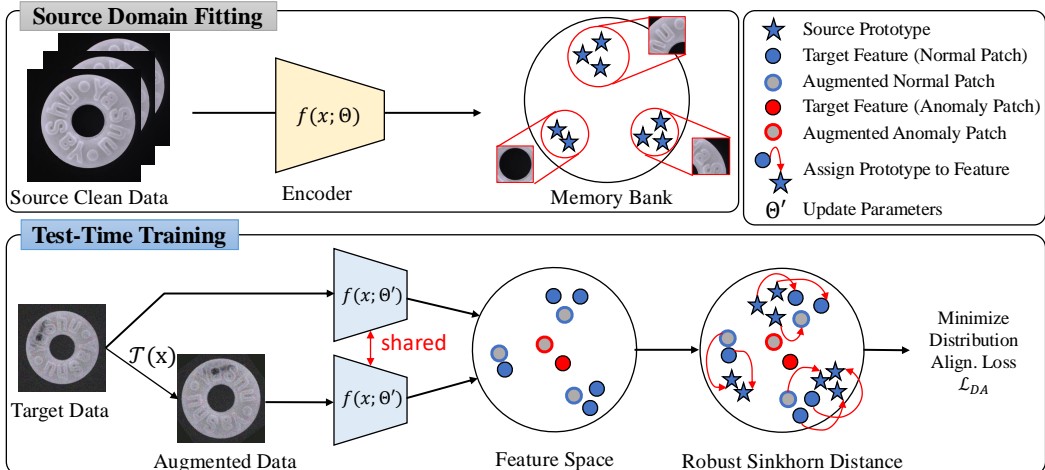

Figure 1: Illustration of pipeline of TTAD. The source domain fitting stage constructs a memory bank of normal training features, which serves as a reference for anomaly detection. In the test-time training stage, target data are augmented and aligned with the source memory bank through robust optimal transport.

tion: i) no retraining or modification of the training process, and ii no access to training data during inference.

To address these constraints, we build upon memory bank-based anomaly detection methods, which have shown impressive performance by explicitly modeling the training data distribution. A notable example, PatchCore Roth et al. (2022), constructs a memory bank of patch-wise image features from normal training samples, capturing the distribution of normal patterns non-parametrically. At inference, testing patches are compared with those in the memory bank for anomaly detection. However, under domain shift, we observe a significant performance drop, attributed to the mismatch between the memory bank and test samples, as illustrated in Fig. 2. This distribution mismatch increases the anomaly score for all testing samples, diminishing the ability to distinguish between normal and anomalous patches. To mitigate this, we propose a test-time training method that adapts to target data distribution during inference.

Recent test-time domain adaptation methods Su et al. (2022); Liu et al. (2021) have addressed distribution alignment for classification tasks. These methods model both source and target domains with parameterized distributions, such as Gaussian or mixtures of Gaussians, and minimize the discrepancy using loss functions like KL-Divergence Su et al. (2022; 2024) or moment-based distances Liu et al. (2021). However, directly applying these objectives in anomaly detection is suboptimal. A single Gaussian distribution may underfit the data Liu et al. (2021), and while mixtures of Gaussians offer more flexibility, their KL-Divergence lacks a closed-form solution, making them unsuitable for test-time training.

Instead, inspired by robust distribution alignment techniques from generative modeling Adler & Lunz (2018) and domain adaptation Courty et al. (2016), we formulate test-time training as an optimal transport problem. This formulation poses two challenges: i) computational efficiency, as the memory bank can contain thousands of samples, requiring a scalable solution, and ii) robustness, as the target domain may include anomalous patches. To address these challenges, we enhance the Sinkhorn distance Cuturi (2013) by discretizing the assignment process and augmenting the target domain data. These improvements lead to more robust distribution alignment, enabling better generalization of pre-trained anomaly detection models. We refer to the final method Test-Time Anomaly Detection (**TTAD**) following the strategy of update encoder network at test-time. An overview of TTAD is presented in Fig. 1.

Our contributions are summarized as follows.

- We identify the challenge of generalization to out-of-distribution testing in industrial anomaly detection and introduce a distribution alignment paradigm to improve generalization at inference.

- We enhance optimal transport-based distribution alignment for anomaly detection by discretizing the assignment process and augmenting the target domain data.

- We establish an extensive benchmark for distribution-shifted industrial anomaly detection, comparing our approach with state-of-the-art methods.

## 2 RELATED WORKS

**Anomaly Detection**: Anomaly Detection (AD) aims to identify samples that deviate significantly from the norm. Mainstream AD approaches primarily focus on unsupervised settings, utilizing various techniques to model normal data Ruff et al. (2018); Yao et al. (2023); Roth et al. (2022); Deng & Li (2022); Xie et al. (2023). One-class classification methods, such as Deep SVDD Ruff et al. (2018), attempt to represent normal data using support vectors. Reconstruction-based methods, like PMAD Yao et al. (2023), train models to recreate normal images and detect anomalies through higher reconstruction errors. Knowledge distillation methods, such as RD4AD Deng & Li (2022), distill normal patterns from pre-trained models and identify anomalies by detecting discrepancies between the distilled and original features. Additionally, distance-based approaches like PatchCore Roth et al. (2022) measure the distance between test image embeddings and reference embeddings from normal training data to detect anomalies. Recently, there has been increasing interest in anomaly detection under distribution shifts during testing. For instance, Cao et al. (2023) builds on reverse distillation techniques Deng & Li (2022), proposing improvements in model generalization by augmenting test data with specific transformations. However, these methods assume that the distribution shift during testing is similar to the augmentations used during training. In contrast, our work addresses a more practical scenario, where the distribution shift at test time differs substantially from training augmentations, and access to normal training samples is not available.

**Domain Adaptation**: Domain adaptation seeks to address the poor generalization caused by distribution shifts between training and testing data. Methods such as learning invariant representations Ganin & Lempitsky (2015) and clustering Tang et al. (2020) have been successful in this area. However, traditional unsupervised domain adaptation approaches require access to both source and target domain data, which is impractical in scenarios where access to source data is restricted due to privacy concerns. This has led to the rise of source-free domain adaptation (SFDA) methods (Liang et al., 2020; Liu et al., 2021; Yang et al., 2021; Liang et al., 2021; Su et al., 2022; 2024), which update models using only target domain data in an unsupervised manner, aiming to improve generalization. Nevertheless, existing SFDA methods are primarily developed for classification tasks, with little consideration for generalizing to anomaly detection. In this work, we adopt a test-time training approach to mitigate distribution shifts by aligning distributions between source and target domains. Specifically, we optimize the optimal transport distance Cuturi (2013) between these distributions. Optimal transport has been widely studied in domain adaptation Courty et al. (2016); Lee et al. (2019) and has been extended to handle outliers Balaji et al. (2020); Mukherjee et al. (2021). Our approach aims to provide a computationally efficient solution that scales well, improving upon the Sinkhorn distance through discretization and target domain augmentation.

**Anomaly Detection under Domain Shift**: Anomaly detection under distribution shift has only recently gained attention Cao et al. (2023). Early attempts to address this challenge involved augmenting data during the training stage to enhance the model's robustness Cao et al. (2023), demonstrating effectiveness in both industrial defect detection and natural OOD (out-of-distribution) images. However, these approaches rely on the assumption that training can be modified and that prior knowledge of the distribution shift is available. In this work, we further relax these assumptions by updating the model only during test time upon observing target data, without modifying the training process. An alternative approach to handling anomaly detection under distribution shifts involves training from scratch using noisy target data Jiang et al. (2022); Chen et al. (2022); McIntosh & Albu (2023). These methods incrementally filter out potential anomalies and learn normal patterns from the remaining clean samples. However, we argue that such methods may struggle to generalize when the noise level in the target distribution is high, limiting their effectiveness in handling severe distribution shifts.

## 3 METHODOLOGY

### 3.1 PROBLEM FORMULATION

We first formally define the task of unsupervised anomaly detection under distribution shift. W.l.o.g, we denote the source domain training data as $\mathcal{D}_s = \{x_i, y_i\}_{i=1\cdots N_s}$ where all samples are defect-free. We further denote the target domain testing data as $\mathcal{D}_t = \{x_j, y_j\}_{j=1\cdots N_t}$ where the labels are not visible. We further denote the distribution from which samples are drawn as $\mathcal{D}_s \sim \mathcal{P}_s$ and $\mathcal{D}_t \sim \mathcal{P}_t$. For anomaly detection purpose, the label only takes a binary value, i.e. $y \in \{0, 1\}$ with 1 indicating anomalous. Following the practice of memory bank based anomaly detection methods Roth et al. (2022), a backbone network $z_i = f(x_i; \Theta) \in \mathcal{R}^{N_p \times D}$ extracts features, as $N_p$ patches, from input sample. A memory bank $\mathcal{M} = \mathcal{C}(\{z_i\}_{i=1\cdots N_s \times N_p}, K)$ takes an abstraction of source domain training samples by sampling a core-set $\mathcal{C}(\cdot, K)$ of size $N_M$ as in Eq. 1. At inference stage, testing sample features are compared against the memory bank to determine anomaly.

$$\min_{\mathcal{M} \in \mathcal{D}_s} \max_{z_j \in \mathcal{D}_s} \min_{z_i \in \mathcal{M}} ||z_i - z_j||, \quad s.t. \quad |\mathcal{M}| \leq N_M \tag{1}$$

The above procedure achieves competitive results for industrial defect identification. Nevertheless, we witness a significant performance drop when testing data experiences a distribution shift, i.e. $p_s \neq p_t$. In this work, we aim to address the distribution shift challenge from a distribution alignment perspective.

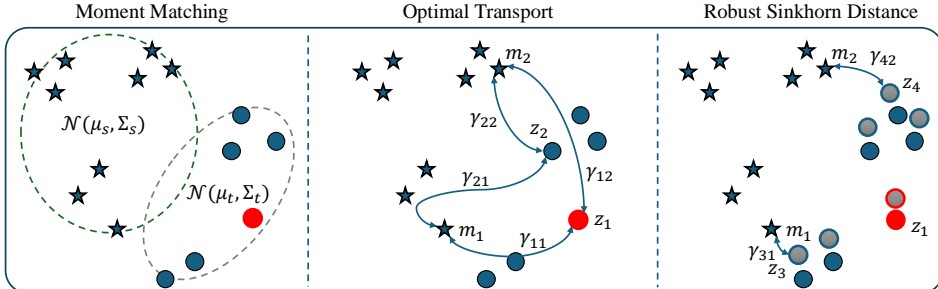

Figure 2: Illustration of distribution alignment via moments matching, optimal transport and finally our modified robust sinkhorn distance.

### 3.2 DISTRIBUTION ALIGNMENT FOR IMPROVING ANOMALY DETECTION

We first identify the underlying reason of why anomaly detection model. After revisiting the mechanism of memory bank based anomaly detection, we notice the anomaly score for each patch is obtained as the shortest distance to any samples in the memory bank.

$$s_i = \max_{p \in 1\cdots N_p} \min_{m_k \in \mathcal{M}} ||z_{ip} - m_k||_2 \tag{2}$$

The above way to characterize anomaly score is built upon the assumption that normal sample distribution is consistent between training and testing data. Therefore, a high anomaly score indicates anomaly. This assumption no longer holds true when distribution shift exists as the overall distance between testing patches and memory bank patches are increased, thus diminishing the discriminability between normal and anomalies.

To mitigate the distribution shift, recent works on test-time domain adaptation proposed distribution alignment approaches Su et al. (2022); Liu et al. (2021). The key insights derived suggest that minimizing the distribution discrepancy between the overall feature distribution of source and target domains could substantially improve the generalization capability. Specifically, a parametric distribution, e.g. multi-variate Gaussian Liu et al. (2021) or mixture of Gaussian Su et al. (2022), is fitted on both source and target domain, denoted as $p_s(z)$ and $p_t(z)$. A loss function that measures the discrepancy between $p_s(z)$ and $p_t(z)$ is employed. For example, Su et al. (2022) introduced

the KL-Divergence between the two Gaussian distributions for alignment as follows. A closed-form solution exists and serves as the loss function to optimize upon target domain data.

$$\mathcal{L}_{DA} = D_{KL}(p_s||p_t) = D_{KL}(\mathcal{N}(\mu_s, \Sigma_s)||\mathcal{N}(\mu_t, \Sigma_t)) \tag{3}$$

Despite the great success in improving the generalization for classification tasks, we argue that such a vanilla distribution alignment approach is sub-optimal for memory bank based anomaly detection task due to the following reason. Without knowing the prior information of the distribution, fitting the model with a single multi-variate Gaussian distribution is prone to underfitting. A mixture of Gaussian may better fit the complex distribution, however, unluckily there is no closed-form solution to the KL-divergence between two mixture of Gaussians Hershey & Olsen (2007). Finally, when the distribution overlap is too small, the gradient of KL-Divergence may be too small, prohibiting gradient-based optimization. Given the above challenge, we resort to a more stable solution to distribution alignment via optimal transport.

### 3.3 DISTRIBUTION ALIGNMENT VIA OPTIMAL TRANSPORT

Inspired by the success of distribution based via optimal transport for unsupervised domain adaptation Courty et al. (2017); Damodaran et al. (2018), we propose to use optimal transport (OT) distance for distribution alignment between $\mathcal{M}$ and $\mathcal{D}_t$. Specifically, a cost matrix $C \in \mathbb{R}^{N_{tp} \times N_M}$ is built between target domain patches and memory bank with $C_{ij} = ||z_i - m_j||$ and $N_{tp} = N_t \cdot N_p$. Assuming uniform weight applied to each sample, the optimal transport is formulated as,

$$\min_{\gamma \geq 0} \sum_i^{N_{tp}} \sum_j^{N_M} \gamma_{ij} C_{ij}, \quad s.t. \quad \sum_i \gamma_{ij} = \frac{1}{N_M}, \ \sum_j \gamma_{ij} = \frac{1}{N_{tp}} \tag{4}$$

Solving the above problem, through linear programming, is expensive and an efficient algorithm, Sinkhorn distance Cuturi (2013), exists that can substantially reduce the computation cost. Specifically, an entropy regularization term is added, giving rise to the following problem. An iterative algorithm is employed to solve the problem.

$$\min_{\gamma \geq 0} \sum_i^{N_{tp}} \sum_j^{N_M} \gamma_{ij} C_{ij} + \epsilon \sum_i \sum_j \gamma_{ij} \log \gamma_{ij}, \quad s.t. \quad \sum_i \gamma_{ij} = \frac{1}{N_M}, \ \sum_j \gamma_{ij} = \frac{1}{N_{tp}} \tag{5}$$

**Self-Training Perspective**: We further elaborate the distribution alignment from a self-training (ST) perspective. ST has been demonstrated to be effective for test-time adaptation Su et al. (2024). The regular routine makes predictions on testing samples and use most confident ones, a.k.a. pseudo labels, to train network, e.g. optimize cross-entropy loss for classification task. In the realm of anomaly detection, self-training could translate into encouraging testing patch to be close to the closest patch in the memory bank. Distribution alignment via optimal transport can be seen as discovering a global optimal assignment between target patches and memory bank. The assignment can be seen as the pseudo label and minimizng the Wasserstein distance is equivalent to using the pseudo label for self-training.

### 3.4 ROBUST SINKHORN DISTANCE

Solving the optimal transport problem in Eq. 5 yields the assignment $\gamma^*$ for each target sample to source samples. The Sinkhorn distance, $\mathcal{L}_{DA} = \sum_i^{N_{tp}} \sum_j^{N_M} \gamma_{ij}^* C_{ij}$, could be adopted as the objective to optimize for distribution alignment. However, we notice a unresolved issue by directly optimizing the above objective. First, an anomalous patch, indexed by $j^*$, in the target domain are always assigned to source patches in the memory bank due to the constraint $\gamma_{ij^*} \geq 0$, $\sum_i \gamma_{ij^*} = \frac{1}{N_M}$. Minimizing the distance between anomalous patches and memory bank patches will inevitably diminish the discriminability. To improve the robust of optimal transport for distribution alignment, we convert the continuous optimal transport assignment into discrete assignment. Fortunately, the discretization may eliminate weak assignments that often appear on anomalous patches in the target

domain. Specifically, we follow the rules below to discretize the assignment, resulting in a more robust distribution alignment loss in Eq. 6. We demonstrate that the above discretization could substantially reduce the overall assignment between anomalous patches and memory bank patches.

$$\mathcal{L}_{DA} = \sum_i^{N_{tp}} \sum_j^{N_M} \pi_{ij}^* C_{ij}, \quad s.t. \quad \pi_{ij}^* = \begin{cases} 1 & \text{if } j = \arg\max_j \gamma_{ij}^*, \text{ or } i = \arg\max_i \gamma_{ij}^* \\ 0 & \text{otherwise} \end{cases} \tag{6}$$

**Target Domain Data Augmentation**: We apply a batchwise update strategy to facilitate gradient based update of backbone weights. Wihin each minibatch we further apply data augmentation $\mathcal{T}(x)$ on the target domain data to improve the distribution alignment, $\tilde{\mathcal{D}}_t = \{\mathcal{T}(x_j)\}_{j=1\cdots N_t}$. The augmentation simulates the normal data variation, e.g. rotation in multiple of $90°$, contrast, etc. Importantly, the augmentation is agnostic to the corruption (distribution shift) on the target domain and is only applied at testing stage, in contrast to the training stage augmentation adopted in GNL Cao et al. (2023). We attribute the effectiveness of test-time target domain augmentation to the following reasons. First, the augmentation will create a more diverse and smoother distribution. This can help mitigate the impact of outliers by "diluting" their influence, making the alignment focus on general features rather than outlier-specific characteristics. Moreover, data augmentation can help by incorporating additional noise into the training process in a controlled way, making the model more resilient to noise and outliers in the real world. The positive effect is demonstrated by the reduced discretised assignment.

## 3.5 OVERALL ALGORITHM

Following the practice of common test-time training strategies, we update the batchnorm affine parameters, $\Theta_{bn}$, with distribution alignment loss. We present the overall algorithm of the proposed method in Algo. 1.

---

**Algorithm 1** Test-Time Training for Anomaly Detection

---

1: **Input:** Pretrained memory bank $\mathcal{M}$, target data $\mathcal{D}_t$, initial encoder network $\Theta$
2: **Output:** Anomaly scores $\{s_i\}$
      # Test-time training on target data
3: **for** $\mathcal{B}_t \subset \mathcal{D}_t$ **do** # Collect one minibatch $\mathcal{B}_t$
         Augment target minibatch $\tilde{\mathcal{B}}_t = \mathcal{T}(\mathcal{B}_t)$
         Compute cost matrix $C \in \mathbb{R}^{|\mathcal{D}_s| \times |\tilde{\mathcal{B}}_t|}$
         Solve optimal transport plan $\gamma^*$ by Eq. 5
         Discretize assignment $\pi^*$ by Eq. 6
         Update model $\Theta_{bn} = \Theta_{bn} - \alpha \frac{\nabla \mathcal{L}_{DA}}{\Theta_{bn}}$
4: **end for**
      # Evaluate on target data
5: **for** $x_i \in \mathcal{D}_t$ **do**
         Encode feature with updated model $z_i = f(x_i; \Theta^*)$
         Per sample anomaly score $s_i = \max_{p \in 1 \cdots N_p} \min_{m_k \in \mathcal{M}} ||z_{ip} - m_k||_2$
6: **end for**
7: **return** $s_i$

---

# 4 EXPERIMENTS

## 4.1 EXPERIMENT SETTINGS

**Dataset**: We evaluate our method on two widely-used 2D industrial anomaly detection datasets, **MVTec** Bergmann et al. (2019) and **RealIAD** Wang et al. (2024), as well as on a 3D dataset, **MVTec 3D** Bergmann et al. (2021).

**MVTec** is the most commonly used benchmark for 2D industrial anomaly detection, comprising 15 object categories, with 60-300 normal samples for training and 30-400 normal and anomalous samples for testing. **RealIAD** is a newly introduced industrial dataset with 30 object categories, each captured from five different viewpoints. We follow the single-view experiment setup, utilizing only the top-view images. Due to the high resolution of the original images (over 3,000×5,000 pixels), which imposes significant computational demands, we use a downsampled version with a resolution of 1,024×1,024. Illustration of the two 2D dataset is shown in 3. **MVTec 3D** consists of 3D scans that include both geometric surface data and RGB information. The dataset comprises 10

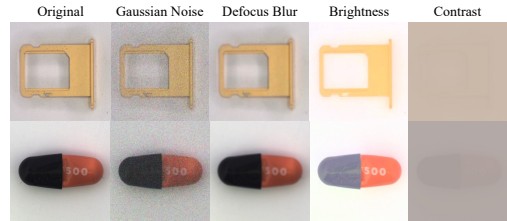

Figure 3: Illustrations of the synthesized distribution shift on MVTec and RealIAD dataset. The severity level of all corruptions is set to 5. It can be observed that the difficulty varies significantly across different corruptions, with contrast being the most challenging. More examples are shown in Appendix Fig. 6 and Fig. 7.

object categories, with over 200 normal images for training and more than 100 images for testing per category.

**Evaluation Protocol**: We simulate commonly seen distribution shift to evaluate the generalization robustness. For the 2D datasets, MVTec and RealIAD, we follow the corruption generation process described in Hendrycks & Dietterich (2019), applying four common corruptions, including Gaussian Noise, Defocus Blur, Contrast, and Brightness, with a severity level of 5 to create distribution-shifted data. In the 3D dataset, MVTec 3D, we simulate natural distribution shifts by randomly adding Gaussian noise $n \sim \mathcal{N}(0, [1e-6]^2)$ to the images. For evaluation, we assess performance using the area under the ROC curve (AUROC), treating anomalies as the positive class for both anomaly detection and segmentation tasks, following the standard protocol Bergmann et al. (2019).

For the 2D experiments, we use a WRN-50 Zagoruyko & Komodakis (2017) pretrained on ImageNet Deng et al. (2009) as the backbone, and only fine-tune the BatchNorm parameters during adaptation. For the 3D experiments, we adopt a PointTransformer Zhao et al. (2021) pretrained on ShapeNet Chang et al. (2015) as the backbone. The batch size during adaptation for all experiments is set to 10, with two types of random geometric augmentations (Flipping and Rotation) applied to each sample. We train for 10, 30, and 1 epochs on the MVTec, MVTec 3D, and RealIAD datasets, respectively. To be noted, the number of epochs is determined based on the complexity and size of each dataset. Due to the large size of the RealIAD dataset, we found that one pass of the data was sufficient for the model to converge. The learning rate is set to 0.003, and the model is optimized using SGD Ruder (2017) with momentum of 0.9.

**Competing Methods**: We compare against several baseline methods, covering several state-of-art industrial AD methods, and three domain adaptation AD methods. These 2D industrial AD methods including reconstruction-based approaches (**ViTAD** Zhang et al. (2023a)), embedding-based methods (**CFLOW-AD** Gudovskiy et al. (2022)), and knowledge distillation methods (**KDAD** Salehi et al. (2021) and **RD4AD** Deng & Li (2022)). We also evaluated on a unified model (**UnIAD** You et al. (2022)), and the effective memory-bank-based method (**PatchCore** Roth et al. (2022)). We further adapted test-time training methods for anomaly detection task. In specific, we evaluated two distribution alignment based test-time training approaches, **TTT++** Liu et al. and **TTAC** Su et al. (2022), on top of PatchCore. Additionally, the state-of-the-art domain adaptation method for anomaly detection, **GNL** Cao et al. (2023), are benchmarked. We allow GNL to re-train with default data augmentation. For 3D anomaly detection, we also benchmark several hand-crafted features implemented by Bergmann et al. (2021), FPFH Horwitz & Hoshen (2022) and M3DM Wang et al. (2023). Finally, we evaluate our proposed method, **TTAD**, on all datasets.

### 4.2 TEST-TIME TRAINING FOR ANOMALY DETECTION

We first present the anomaly detection results, averaged across all object classes, on both the MVTec and RealIAD datasets in Table 1. A more detailed results for per-class AUROC are deferred to the Appendix. From the results, we derive the following key observations:

**i)** State-of-the-art anomaly detection methods struggle significantly under distribution shifts, as evidenced by the performance gap when tested on clean versus corrupted target data. For instance, PatchCore shows a performance drop of 21.46% when exposed to Gaussian noise on the MVTec dataset. This highlights the vulnerability of these methods to out-of-distribution (OOD) scenarios. **ii)** Test-time training methods (e.g., TTT++ and TTAC), despite showing strong performance on classification tasks, fail to deliver comparable results on anomaly detection tasks. In fact, both TTT++ and TTAC underperform in most cases compared to PatchCore, which performs no adaptation. This underperformance can be attributed to the methods' reliance on modeling complex distributions with a single Gaussian distribution, leading to underfitting. **iii)** In contrast, TTAD, which leverages distribution alignment via optimal transport, demonstrates superior performance in 3 out of 4 types of corruptions, with the sole exception being the "Contrast" corruption. Notably, under the Defocus Blur and Brightness corruptions on the MVTec dataset, TTAD's performance is only 2% behind the results on source domain. These findings underscore the importance of a well-calibrated distribution strategy for robust anomaly detection. **iv)** Lastly, we observe that GNL significantly outperforms all competing methods under the "Contrast" corruption scenario. Upon further investigation, we discovered that GNL employs an "AutoContrast" augmentation during training, which inadvertently provides prior knowledge of the target data distribution. This unfair advantage highlights the importance of evaluating methods under consistent and unbiased conditions.

Table 1: Results of anomaly detection on MVTec and RealIAD datasets. We report the mAU-ROC(%) averaged across all classes. "Clean" refers to the results on clean testing samples.

| | MVTec | | | | | RealIAD | | | | |
|---|---|---|---|---|---|---|---|---|---|---|
| | Clean | Gauss. Noise | Defoc. Blur | Bright. | Contrast | Clean | Gauss. Noise | Defoc. Blur | Bright. | Contrast |
| ViTAD | 98.30 | 63.86 | 79.82 | 67.20 | 53.61 | 82.70 | 52.43 | 73.43 | 61.15 | 57.43 |
| KDAD | 87.74 | 74.44 | 78.79 | 72.67 | 44.05 | 80.23 | 41.15 | 31.24 | 38.31 | 46.65 |
| RD4AD | 98.50 | 81.03 | 93.00 | 90.27 | 65.08 | 86.17 | 56.57 | 79.54 | 63.73 | 57.42 |
| UnIAD | 92.50 | 84.05 | 79.83 | 90.03 | 61.29 | 83.10 | 64.17 | 78.84 | 69.44 | 53.95 |
| CFLOW-AD | 91.55 | 59.52 | 60.54 | 59.71 | 51.50 | 77.00 | 56.01 | 62.57 | 56.47 | 53.18 |
| PatchCore | 98.81 | 77.34 | 90.43 | 91.19 | 62.72 | 90.35 | 60.24 | 77.02 | 63.01 | 50.36 |
| TTT+ | 98.81 | 71.82 | 71.30 | 76.37 | 70.30 | 90.35 | 52.07 | 50.69 | 44.64 | 50.73 |
| TTAC | 98.81 | 56.41 | 82.88 | 55.34 | 55.34 | 90.35 | 53.93 | 60.55 | 54.74 | 53.15 |
| GNL | 97.99 | 83.75 | 95.27 | 92.96 | **88.20** | 83.44 | 62.72 | 79.57 | 64.51 | **62.28** |
| TTAD (**Ours**) | 98.81 | **89.21** | **96.75** | **96.71** | 85.45 | 90.35 | **69.73** | **83.29** | **69.55** | 60.71 |

In addition to the experiments on 2D data, we also evaluated our method on 3D data by introducing Gaussian noise as a form of corruption. The results are presented in Table 2. Compared to standard 2D corruptions, adding Gaussian noise to 3D data introduces a greater challenge. The geometric structures in 3D data are particularly sensitive to noise, as it disrupts fine details and depth information, both of which are crucial for effective 3D anomaly detection. Despite these challenges, our method demonstrates resilience, achieving a notable performance improvement of 1.5% on PointMAE. This suggests that our approach is capable of effectively managing the complexities introduced by noise in 3D data, maintaining robust anomaly detection capabilities.

Table 2: Results of anomaly detection on MVTec-3D with per class AUROC(%)

| | Bagel | CableGland | Carrot | Cookie | Dowel | Foam | Peach | Potato | Rope | Tire | Mean |
|---|---|---|---|---|---|---|---|---|---|---|---|
| Depth GAN | 47.5 | 24.0 | 49.1 | 45.9 | 37.4 | 36.8 | 32.4 | 37.0 | 35.1 | 36.5 | 38.17 |
| Depth AE | 33.4 | 38.6 | 43.3 | 47.9 | 40.7 | 32.3 | 42.9 | 41.6 | 41.2 | 38.3 | 40.02 |
| Depth VM | 36.7 | 32.2 | 37.4 | 44.6 | 40.4 | 29.2 | 38.7 | 29.5 | 45.3 | 39.7 | 37.37 |
| Depth PatchCore | 75.8 | 53.8 | 64.3 | 75.5 | 44.6 | 48.4 | 40.8 | 50.7 | 56.5 | 56.6 | 56.70 |
| Raw (in BTF) | 58.4 | 49.8 | 44.8 | 45.7 | 50.2 | 33.2 | 24.7 | 31.1 | 44.6 | 50.4 | 43.29 |
| HoG (in BTF) | 61.2 | 57.2 | 33.0 | 56.9 | 51.1 | 41.8 | 38.4 | 69.2 | 50.0 | 60.6 | 51.94 |
| SIFT (in BTF) | 46.1 | 42.3 | 44.1 | 46.6 | 38.5 | 41.9 | 33.4 | 55.7 | 62.4 | 56.4 | 46.74 |
| FPFH | 49.4 | 48.0 | 54.8 | 37.0 | 38.8 | 38.7 | 36.5 | 50.7 | 51.9 | 49.8 | 45.56 |
| M3DM | 74.1 | 51.6 | 73.2 | 83.2 | 59.9 | 58.6 | 30.0 | 76.3 | 86.8 | 70.8 | 66.45 |
| TTAD (**Ours**) | 80.2 | 58.6 | 73.4 | 86.4 | 60.6 | 51.8 | 46.6 | 80.2 | 83.3 | 58.5 | **67.96** |

### 4.3 TEST-TIME TRAINING FOR ANOMALY SEGMENTATION

We also evaluate the anomaly segmentation performance on the MVTec and RealIAD dataset, with the results summarized in Table 3. Detailed results for each class are deferred to the Appendix. For a

fair comparison, we include only those methods that provide segmentation solutions in their original papers. As shown in the table, our method consistently achieves superior AUROC across all types of corruption in the segmentation task. Notably, it surpasses all baseline methods across different corruptions on MVTec dataset. While RD4AD also performs well under Defocus Blur, our method maintains an advantage. Moreover, under Brightness and Gaussian Noise corruptions, our approach outperforms RD4AD by significant margins of 7.25% and 5.61%, respectively. On the RealIAD dataset, our method slightly lags behind UnIAD under Brightness, while showing a significant lead in the other three corruption types.

Table 3: Anomaly segmentation results on MVTec and RealIAD datasets. P-mAUROC(%) across all classes.

| | MVTec | | | | | RealIAD | | | | |
|---|---|---|---|---|---|---|---|---|---|---|
| | Clean | Gauss. Noise | Defoc. Blur | Brightness | Contrast | Clean | Gauss. Noise | Defoc. Blur | Brightness | Contrast |
| CFLOW-AD | 95.65 | 70.38 | 79.19 | 75.87 | 50.02 | 88.60 | 71.24 | 91.05 | 85.14 | 72.05 |
| UnIAD | 95.70 | 70.02 | 87.04 | 90.86 | 72.72 | 86.00 | 87.95 | 96.26 | **90.15** | 80.68 |
| RD4AD | 97.80 | 86.88 | 96.52 | 90.83 | 78.68 | 89.22 | 56.52 | 95.64 | 76.99 | 83.08 |
| PatchCore | 98.34 | 87.00 | 93.34 | 91.39 | 71.07 | 98.10 | 76.99 | 96.63 | 83.49 | 74.49 |
| TTT++ | 98.34 | 80.31 | 79.21 | 79.20 | 78.86 | 98.10 | 59.33 | 46.15 | 44.64 | 47.30 |
| TTAC | 98.34 | 56.77 | 38.43 | 82.28 | 52.59 | 98.10 | 52.93 | 63.37 | 62.24 | 50.30 |
| Ours | 98.34 | **94.13** | **96.53** | **96.44** | **90.54** | 98.10 | **89.95** | **98.47** | 89.02 | **83.30** |

**Qualitative Results**: We provide a qualitative comparison of anomaly segmentation results, as shown in Fig. 4. We compare our method with Patchcore without adaptation and the second-best overall baseline, RD4AD. The source predictions serve as reference upperbound. RD4AD performs well under simpler corruptions like Defocus Blur, achieving relatively accurate anomaly localization. However, under more challenging corruptions such as Brightness and Contrast, it tends to misidentify the entire background or object as the anomaly area. In contrast, our method shows a significant improvement compared to the no-adaptation model, which lacks segmentation capability,

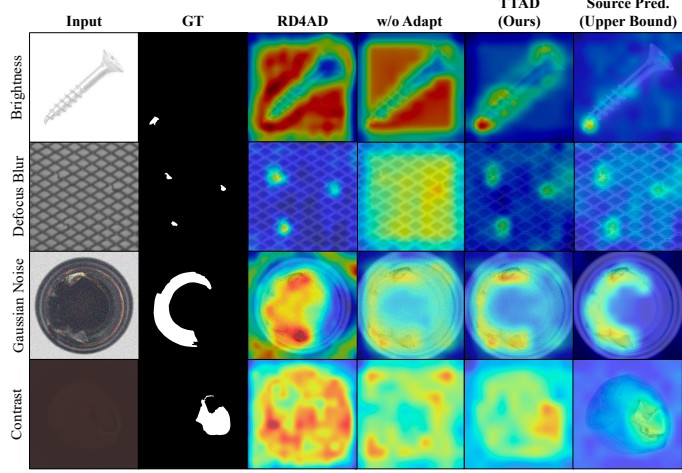

Figure 4: Qualitative results for anomaly segmentation. We present results for PatchCore without adaptation (w/o Adapt), RD4AD, TTAD (Ours) and predictions on clean testing sample as upperbound (Source Pred.). TTAD consistently improves anomaly localization compared to the baseline (w/o Adapt), sometimes even approaching the upperbound.

and demonstrate a strong and consistent performance which is closely approaching the upper bound results on clean samples.

## 4.4 ABLATION STUDY

We analyze the effectiveness of proposed methods by investigating distribution alignment method, assignment method and target data augmentation. The ablation study carried out on MVTec dataset is presented in Tab. 4. We make the following observations from the results. **i)** KL-Div Su et al. (2024) and Moment Matching Liu et al. (2021)-based alignment exhibit the poorest performance across all corruption types, with particularly low scores in the pixel-wise AUROC, especially under Defocus Blur (e.g., 38.43% for Moment Matching) and Gaussian Noise (56.77% for Moment Matching). This indicates that these alignment methods are not well-suited for handling complex distribution shifts in anomaly detection tasks. In contrast, optimal Transport-based alignment consistently outperforms KL-Div and Moment Matching across all corruptions. **ii)** We further compared with another way of discrete optimal transport solution, i.e. using Hungarian Method to find linear assignment between memory bank and target samples. Introducing Hungary Method assignment

for Optimal Transport yields better results compared to no assignment strategy, as seen in the case of Gaussian Noise. Similar improvements are observed across other corruptions like Defocus Blur (94.81% vs. 93.31%) and Contrast (75.73% vs. 71.51%). **iii)** When directly copying the target data for augmentation, the performance improves further, particularly in pixel-wise AUROC. Applying data augmentation instead of direct copying results in the best performance overall.

In summary, the best-performing configuration combines Optimal Transport alignment with discrete assignment and data augmentation, achieving top scores in both instance-level and pixel-wise AUROC across all corruption types. Notably, the Contrast corruption is still posing great challenge to the method which is explained by the low visibility of defects.In contrast, the KL-Div and Moment Matching methods consistently underperform, indicating that more sophisticated distribution alignment techniques, like Optimal Transport, are critical for handling complex distribution shifts in anomaly detection tasks.

Table 4: Ablation study on MVTec dataset. We report anomaly detection and segmentation AUROC averaged over all classes (mAUROC & P-mAUROC).

| Distribution Alignment | Assignment | Target Data Aug. | Gaussian Noise | | Defocus Blur | | Brightness | | Contrast | |
|---|---|---|---|---|---|---|---|---|---|---|
| | | | mAUROC | P-mAUROC | mAUROC | P-mAUROC | mAUROC | P-mAUROC | mAUROC | P-mAUROC |
| - | - | - | 77.34 | 87.00 | 90.43 | 92.34 | 91.19 | 91.40 | 62.72 | 71.07 |
| KL-Div | - | - | 56.41 | 56.77 | 58.27 | 38.43 | 82.87 | 82.28 | 55.34 | 52.59 |
| Moment Matching | - | - | 71.82 | 80.31 | 71.30 | 79.21 | 76.16 | 79.20 | 70.30 | 78.86 |
| OptimalTransport | Hungary Method | - | 83.85 | 89.31 | 93.32 | 96.19 | 93.82 | 94.67 | 71.51 | 80.56 |
| OptimalTransport | discrete | - | 85.72 | 93.36 | 94.81 | 96.41 | 94.72 | 96.32 | 75.73 | 85.45 |
| OptimalTransport | discrete | direct copy | 85.76 | 93.33 | 94.83 | 96.44 | 94.69 | 96.30 | 75.79 | 85.51 |
| OptimalTransport | discrete | data augment. | 89.21 | 94.07 | 96.75 | 97.45 | 96.71 | 96.99 | 84.45 | 90.45 |

**Target Data Augmentation**: We further demonstrate the effectiveness of the proposed augmentation method by analyzing the anomaly sample assignments in the optimal transport solutions. As illustrated in Figure 5, out of 2,090 samples in the target domain memory bank, the number of assignments to anomaly samples significantly decreased from 123 (5.89%) to 79 (3.78%) when applying our data augmentation strategy. This reduction highlights the method's ability to limit erroneous anomaly assignments, thereby enhancing the quality of the optimal transport solution.

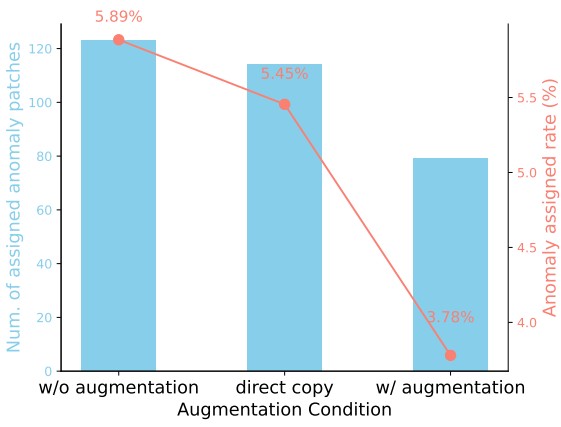

Additionally, we evaluate the impact of simply duplicating the target data for augmentation, which led to a slight reduction

Figure 5: Comparison of anomaly sample assignments across different augmentation strategies.

in anomaly assignments to 114 (5.45%). We attribute this minor improvement to the larger selection pool, though this approach fails to smooth the distribution effectively.

## 5 CONCLUSION

In this work, we addressed a realistic challenge of deploying anomaly detection model to out-of-distribution testing data. Existing works require modifying training objective and require access training data during inference. We relaxed these assumptions by proposing a test-time distribution alignment method to mitigate the distribution shift. In particular, a robust Sinkhorn distance is adapted from an existing optimal transport problem to improve the resilience to anomalous patches in the target domain data. We demonstrated the effectiveness on three industrial anomaly detection datasets. The findings suggest future research should pay more attention to the robustness of anomaly detection under realistic challenges.

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

# A APPENDIX

## A.1 ILLUSTRATION OF DATASETS

We further illustrate the MVTec and RealIAD datasets in Figure 6 7. In general, we find the Contrast corruption is most challenging as differentiating the foreground and background becomes even impossible. This also aligns with the observation that all methods yield the worst performance on Contrast corruption.

## A.2 FULL EXPERIMENTAL RESULTS

We present the full detection and segmentation results of MVTec and RealIAD dataset in Table 5 6 7 and Figure A.2.
Our method consistently achieves the highest performance across almost all classes in Noise, Defocus Blur and Brightness corruptions, demonstrating a clear advantage in terms of AUROC scores.

Original  Gaussian Noise  Defocus Blur  Brightness  Contrast

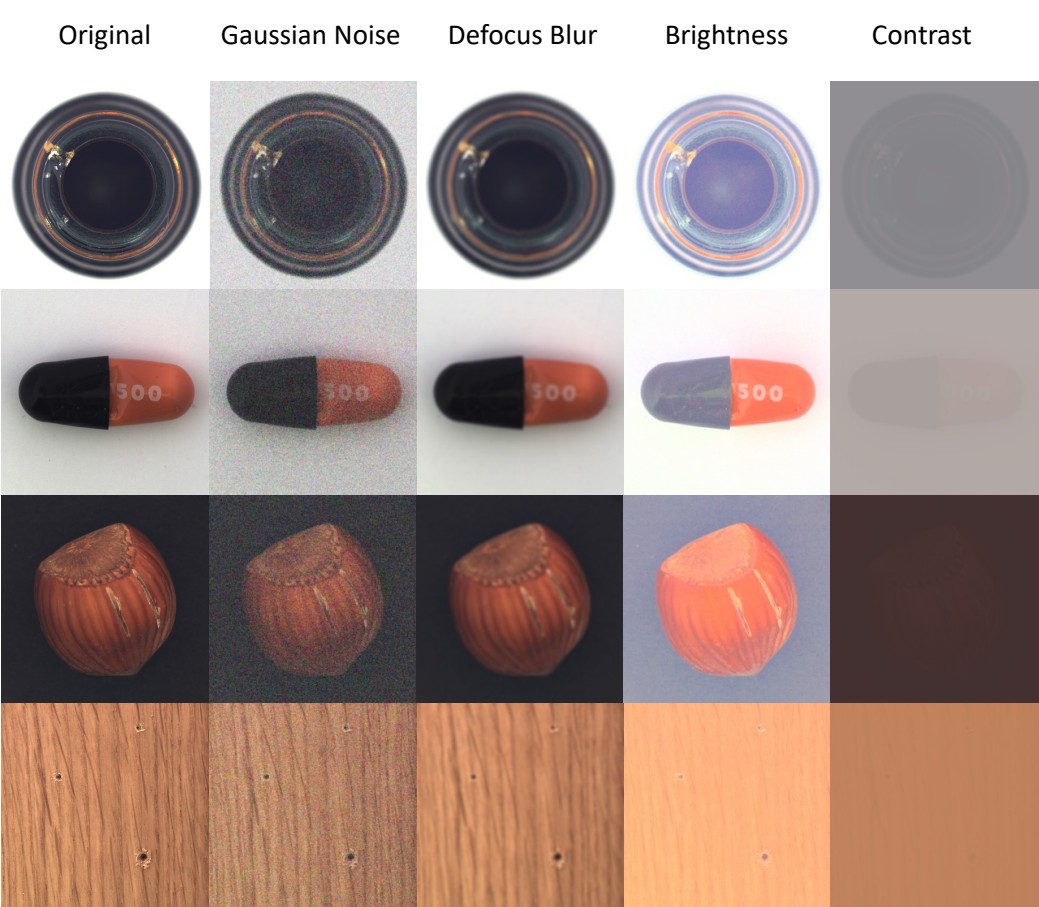

Figure 6: Illustrations of the corruptions on MVTec dataset. Severity level of all corruptions are set to 5.

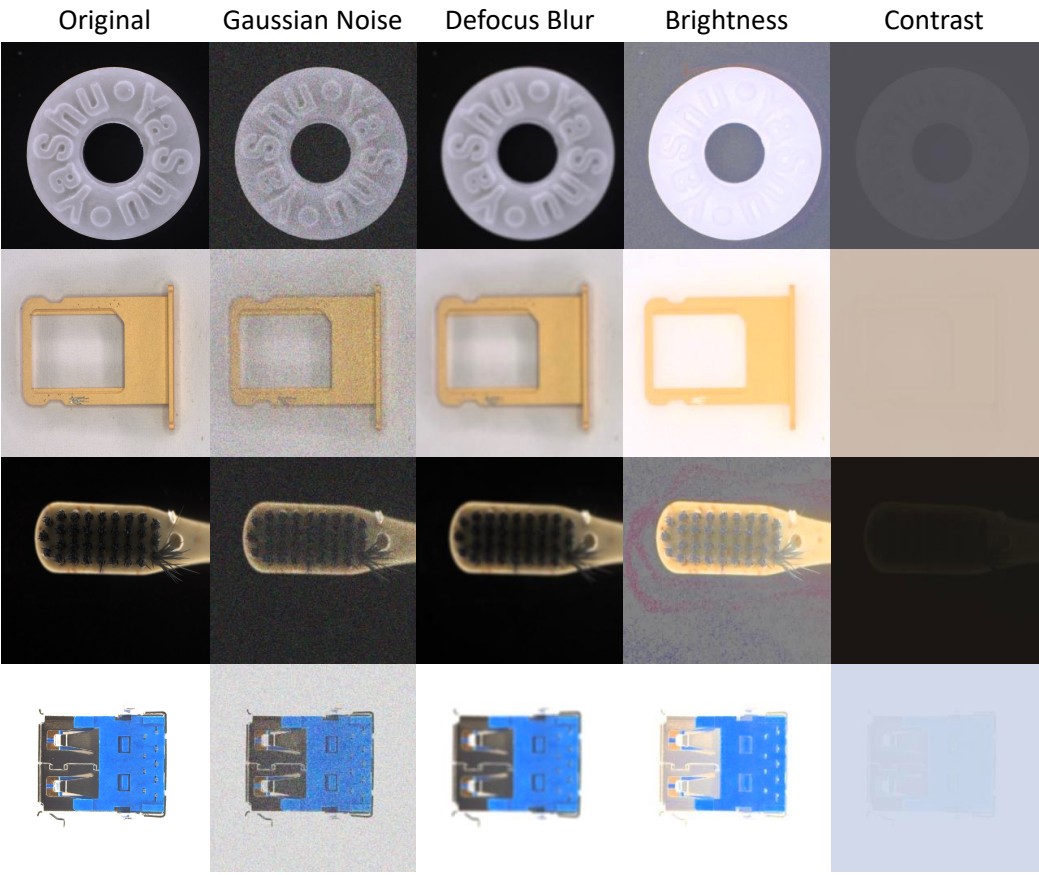

Figure 7: Illustrations of the corruptions on RealIAD dataset. Severity level of all corruptions are set to 5.

Although our method falls behind GNL in a few classes under Contrast, it still maintains competitive results overall. It is worth noting that GNL benefits from its own specialized augmentations, as discussed in the main text. Despite this, our approach continues to deliver robust performance across all other corruption types and remains highly effective in most classes under Contrast.

Table 5: MVTec per class instance AUROC(%)

| Corruption | Method | bottle | cable | capsule | carpet | grid | hazelnut | leather | metal_nut | pill | screw | tile | toothbrush | transistor | wood | zipper | mean |
|---|---|---|---|---|---|---|---|---|---|---|---|---|---|---|---|---|---|
| Gauss. Noise | ViTAD | 63.90 | 66.39 | 82.20 | 82.14 | 74.44 | 69.08 | 71.83 | 30.42 | 60.07 | 61.60 | 68.74 | 53.37 | 55.43 | 68.46 | 49.78 | 63.86 |
| | RD4AD | 54.40 | 97.12 | 59.26 | 95.20 | 89.93 | 95.90 | 99.01 | 74.37 | 64.59 | 64.82 | 89.46 | 73.65 | 88.84 | 96.75 | 72.17 | 81.03 |
| | KDAD | 95.24 | 90.22 | 67.93 | 72.67 | 38.60 | 79.39 | 93.65 | 82.26 | 67.95 | 7.89 | 85.82 | 87.50 | 83.46 | 72.37 | 91.60 | 74.44 |
| | UnIAD | 95.39 | 91.52 | 59.51 | 98.67 | 83.20 | 97.39 | 93.71 | 86.21 | 59.76 | 57.86 | 90.22 | 80.55 | 90.70 | 92.63 | 83.43 | 84.05 |
| | CFLOW-AD | 52.50 | 81.00 | 46.31 | 73.11 | 39.60 | 70.39 | 76.97 | 51.52 | 52.35 | 41.81 | 71.28 | 51.67 | 68.87 | 65.79 | 49.58 | 59.52 |
| | patchcore | 81.59 | 88.87 | 49.06 | 91.49 | 52.13 | 95.11 | 97.28 | 72.97 | 52.02 | 40.44 | 90.98 | 73.61 | 81.92 | 95.18 | 97.45 | 77.34 |
| | TTAC | 84.13 | 30.88 | 48.78 | 53.45 | 57.56 | 51.21 | 50.85 | 58.06 | 46.24 | 50.19 | 45.17 | 83.89 | 68.25 | 69.39 | 48.06 | 56.41 |
| | TTT++ | 84.84 | 71.95 | 50.10 | 86.76 | 52.80 | 72.50 | 93.14 | 72.63 | 50.35 | 38.20 | 88.96 | 72.22 | 83.46 | 95.53 | 63.87 | 71.82 |
| | GNL | 67.06 | 96.21 | 64.58 | 95.26 | 90.14 | 97.79 | 98.95 | 68.77 | 63.64 | 66.12 | 96.39 | 95.00 | 88.12 | 96.93 | 71.27 | 83.75 |
| | **Ours** | 93.33 | 98.18 | 72.00 | 97.51 | 87.89 | 98.36 | 98.13 | 89.59 | 73.95 | 69.22 | 95.78 | 81.11 | 91.54 | 97.02 | 94.54 | **89.21** |
| Defoc. Blur | ViTAD | 74.78 | 59.63 | 90.05 | 71.02 | 80.86 | 95.41 | 95.45 | 82.62 | 81.00 | 67.79 | 89.56 | 63.33 | 66.62 | 84.09 | | 79.83 |
| | RD4AD | 99.93 | 87.04 | 83.26 | 95.48 | 93.21 | 100.00 | 100.00 | 98.90 | 90.63 | 79.52 | 93.51 | 93.60 | 92.62 | 99.01 | 88.25 | 93.00 |
| | KDAD | 98.57 | 82.44 | 74.31 | 54.53 | 42.61 | 92.39 | 97.04 | 78.40 | 72.80 | 53.74 | 91.96 | 85.56 | 88.88 | 76.67 | 91.94 | 78.79 |
| | UnIAD | 99.84 | 93.29 | 76.38 | 97.55 | 93.31 | 99.53 | 100.00 | 96.67 | 84.42 | 92.39 | 99.42 | 94.13 | 99.75 | 91.01 | 98.29 | 94.40 |
| | CFLOW-AD | 50.56 | 69.77 | 53.17 | 62.72 | 59.31 | 92.75 | 53.12 | 71.55 | 56.96 | 55.28 | 69.52 | 50.56 | 76.00 | 41.67 | 45.09 | 60.54 |
| | patchcore | 100 | 89.75 | 82.53 | 96.07 | 74.6 | 99.46 | 100 | 96.53 | 84.04 | 56.18 | 93.54 | 92.22 | 96.33 | 96.75 | 98.5 | 90.43 |
| | TTAC | 48.97 | 49.81 | 62.39 | 55.06 | 54.30 | 46.25 | 67.90 | 46.38 | 35.46 | 99.51 | 60.79 | 51.39 | 58.67 | 61.32 | 75.81 | 58.27 |
| | TTT++ | 87.94 | 81.18 | 45.79 | 89.13 | 62.41 | 73.14 | 92.83 | 64.13 | 37.15 | 42.18 | 93.11 | 72.22 | 82.83 | 93.68 | 51.76 | 71.30 |
| | GNL | 100.00 | 97.19 | 85.12 | 97.91 | 94.99 | 100.00 | 100.00 | 99.85 | 91.08 | 81.16 | 99.71 | 95.83 | 95.62 | 97.54 | 97.91 | 95.27 |
| | **Ours** | 100.00 | 94.88 | 91.18 | 98.31 | 96.16 | 100.00 | 100.00 | 98.83 | 91.08 | 88.15 | 96.90 | 99.72 | 99.17 | 98.07 | 98.74 | **96.75** |
| Brightness | ViTAD | 64.92 | 67.78 | 87.75 | 52.57 | 52.18 | 89.53 | 91.56 | 56.96 | 40.08 | 92.48 | 25.80 | 83.45 | 55.44 | 71.85 | 75.63 | 67.20 |
| | RD4AD | 99.87 | 95.99 | 79.75 | 98.81 | 98.34 | 100.00 | 100.00 | 98.94 | 68.18 | 59.69 | 97.71 | 68.90 | 95.06 | 99.70 | 93.12 | 90.27 |
| | KDAD | 80.16 | 75.71 | 75.27 | 72.11 | 56.81 | 82.86 | 87.70 | 65.64 | 70.57 | 80.49 | 74.64 | 68.89 | 81.54 | 41.58 | 76.02 | 72.67 |
| | UnIAD | 99.76 | 95.65 | 67.41 | 99.59 | 87.63 | 99.35 | 100.00 | 97.31 | 79.48 | 51.30 | 93.39 | 81.66 | 100.00 | 100.00 | 97.95 | 90.03 |
| | CFLOW-AD | 64.68 | 69.15 | 51.85 | 87.58 | 49.21 | 77.93 | 52.92 | 51.47 | 54.94 | 57.41 | 85.35 | 31.11 | 56.37 | 55.00 | 50.63 | 59.71 |
| | patchcore | 100.00 | 94.00 | 84.80 | 95.51 | 96.49 | 98.75 | 97.52 | 98.29 | 71.6 | 67.41 | 92.53 | 78.33 | 95.63 | 98.51 | | 91.19 |
| | TTAC | 100 | 95.16 | 55.96 | 92.58 | 97.74 | 100 | 98.68 | 63.2 | 40.4 | 56.1 | 69.44 | 78.61 | 96.62 | 98.77 | 99.87 | 82.88 |
| | TTT++ | 98.25 | 87.46 | 59.99 | 81.78 | 92.48 | 65.79 | 88.35 | 86.66 | 49.95 | 46.00 | 87.45 | 78.61 | 84.38 | 81.75 | 53.62 | 76.17 |
| | GNL | 99.32 | 100.00 | 97.99 | 98.74 | 99.82 | 100.00 | 100.00 | 95.22 | 81.93 | 63.48 | 95.29 | 99.56 | 85.78 | 84.17 | 93.07 | 92.96 |
| | **Ours** | 100.00 | 98.41 | 92.74 | 97.33 | 100.00 | 100.00 | 99.56 | | | | 93.61 | 99.46 | 99.82 | 99.45 | | **96.71** |
| Contrast | ViTAD | 33.88 | 40.14 | 54.58 | 45.38 | 46.27 | 55.39 | 55.37 | 66.16 | 54.67 | 80.57 | 44.08 | 71.47 | 54.15 | 58.91 | 43.11 | 53.61 |
| | RD4AD | 68.38 | 86.70 | 63.63 | 63.52 | 73.24 | 79.19 | 69.76 | 29.41 | 59.24 | 87.22 | 59.45 | 67.57 | 49.77 | 77.73 | | 65.08 |
| | KDAD | 60.56 | 51.44 | 36.94 | 39.85 | 39.77 | 65.96 | 68.40 | 22.92 | 47.95 | 2.20 | 61.47 | 33.06 | 47.79 | 25.79 | 56.72 | 44.05 |
| | UnIAD | 57.30 | 57.53 | 29.75 | 80.29 | 64.57 | 69.57 | 80.16 | 74.82 | 66.39 | 76.86 | 64.64 | 36.38 | 64.16 | 65.26 | 31.69 | 61.29 |
| | CFLOW-AD | 57.54 | 70.52 | 61.95 | 53.55 | 50.46 | 41.82 | 50.00 | 62.46 | 32.19 | 52.65 | 49.96 | 43.61 | 62.58 | 32.94 | 50.29 | 51.50 |
| | PatchCore | 76.98 | 58.88 | 55.33 | 69.10 | 55.05 | 71.64 | 81.28 | 72.19 | 39.69 | 46.61 | 94.84 | 47.50 | 44.04 | 60.00 | 67.62 | 62.72 |
| | TTAC | 100.00 | 95.16 | 55.96 | 92.58 | 97.74 | 100.00 | 98.68 | 63.20 | 56.10 | 69.44 | 78.61 | 96.62 | 98.77 | 99.87 | | 82.88 |
| | TTT++ | 72.06 | 84.03 | 49.90 | 93.26 | 48.45 | 78.25 | 91.58 | 70.23 | 40.21 | 42.63 | 86.22 | 72.22 | 81.04 | 86.93 | 57.54 | 70.30 |
| | GNL | 71.67 | 95.84 | 78.10 | 96.03 | 86.88 | 99.50 | 97.35 | 99.27 | 74.77 | 63.58 | 99.49 | 95.00 | 81.12 | 96.93 | 87.45 | **88.20** |
| | **Ours** | 93.02 | 92.82 | 74.51 | 89.09 | 80.87 | 98.79 | 87.64 | 87.59 | 55.97 | 73.50 | 92.71 | 76.67 | 86.38 | 92.54 | 84.61 | 84.45 |

Table 6: RealIAD per class instance AUROC(%)

| | Gaussian Noise | | | | | | | | | |
|---|---|---|---|---|---|---|---|---|---|---|
| | KDAD | RD4AD | UnIAD | ViTAD | CFLOW-AD | patchcore | TTT++ | TTAC | GNL | **Ours** |
| audiojack | 55.30 | 61.60 | 78.53 | 51.40 | 51.23 | 67.46 | 50.91 | 48.67 | 71.28 | 79.60 |
| bottle_cap | 41.52 | 52.70 | 69.23 | 51.26 | 54.78 | 52.11 | 54.44 | 50.50 | 54.53 | 57.49 |
| button_battery | 42.19 | 63.40 | 57.70 | 69.86 | 49.78 | 64.52 | 56.20 | 55.74 | 67.51 | 72.07 |
| end_cap | 55.16 | 51.20 | 46.08 | 49.47 | 58.88 | 50.90 | 52.20 | 42.47 | 55.99 | 51.12 |
| eraser | 38.26 | 52.90 | 66.83 | 45.67 | 49.13 | 50.22 | 40.60 | 54.62 | 52.13 | 64.23 |
| fire_hood | 38.03 | 44.50 | 65.08 | 46.19 | 50.72 | 51.30 | 46.94 | 48.24 | 45.78 | 62.55 |
| mint | 51.83 | 47.70 | 43.50 | 51.91 | 50.26 | 54.67 | 52.30 | 49.56 | 52.93 | 52.81 |
| mounts | 42.29 | 56.90 | 65.66 | 52.32 | 58.92 | 58.53 | 55.85 | 49.04 | 61.21 | 70.93 |
| pcb | 33.69 | 59.10 | 47.69 | 47.60 | 53.46 | 66.14 | 46.25 | 55.15 | 75.20 | 72.30 |
| phone_battery | 38.90 | 78.20 | 76.33 | 73.59 | 55.82 | 77.64 | 55.40 | 56.09 | 75.22 | 82.53 |
| plastic_nut | 44.09 | 56.40 | 66.45 | 54.15 | 53.43 | 55.70 | 50.02 | 54.23 | 58.85 | 60.90 |
| plastic_plug | 20.93 | 53.70 | 63.62 | 71.79 | 36.93 | 67.69 | 57.45 | 65.15 | 63.80 | 64.68 |
| porcelain_doll | 51.79 | 57.50 | 62.20 | 51.28 | 53.59 | 55.86 | 48.04 | 52.25 | 54.51 | 71.02 |
| regulator | 45.07 | 52.20 | 48.05 | 49.29 | 49.56 | 48.62 | 45.33 | 52.93 | 52.57 | 54.84 |
| rolled_strip_base | 45.77 | 64.00 | 66.33 | 52.16 | 54.23 | 63.07 | 62.02 | 68.15 | 66.20 | 66.70 |
| sim_card_set | 47.44 | 67.40 | 76.45 | 38.44 | 55.91 | 74.79 | 64.39 | 72.73 | 75.99 | 80.50 |
| switch | 26.89 | 57.60 | 57.22 | 53.39 | 70.14 | 64.83 | 57.04 | 51.51 | 69.52 | 72.22 |
| tape | 49.34 | 48.40 | 72.89 | 45.95 | 62.03 | 58.55 | 46.55 | 51.36 | 60.88 | 79.55 |
| terminalblock | 35.64 | 53.10 | 45.14 | 46.32 | 55.72 | 61.83 | 56.15 | 61.26 | 54.28 | 68.40 |
| toothbrush | 29.12 | 57.30 | 66.21 | 64.74 | 65.67 | 71.11 | 46.03 | 61.69 | 70.96 | 82.34 |
| toy | 50.13 | 48.70 | 61.41 | 46.73 | 51.97 | 47.95 | 53.22 | 51.81 | 47.96 | 52.50 |
| toy_brick | 34.26 | 41.70 | 75.04 | 53.54 | 63.35 | 72.57 | 48.64 | 69.93 | 51.05 | 79.04 |
| transistor1 | 29.92 | 56.30 | 70.96 | 68.11 | 50.52 | 47.36 | 39.91 | 48.49 | 71.43 | 86.29 |
| u_block | 39.24 | 50.80 | 63.54 | 53.18 | 57.00 | 57.91 | 50.21 | 52.76 | 56.77 | 70.24 |
| usb | 30.93 | 63.00 | 64.82 | 53.69 | 69.61 | 64.54 | 47.68 | 35.22 | 79.52 | 78.52 |
| usb_adaptor | 41.02 | 55.40 | 65.14 | 49.09 | 51.84 | 62.03 | 54.06 | 52.43 | 54.52 | 62.80 |
| vcpill | 64.80 | 60.80 | 74.73 | 24.09 | 56.10 | 55.16 | 56.98 | 46.50 | 74.77 | 77.17 |
| wooden_beads | 39.67 | 61.60 | 66.80 | 55.29 | 64.42 | 45.06 | 50.61 | 52.09 | 61.04 | 54.99 |
| woodstick | 45.02 | 53.70 | 68.05 | 47.91 | 53.96 | 61.70 | 54.63 | 60.52 | 56.68 | 64.92 |
| zipper | 26.15 | 69.20 | 73.29 | 54.51 | 71.23 | 77.23 | 62.18 | 46.84 | 88.42 | 98.79 |
| mean | 41.15 | 56.57 | 64.17 | 52.43 | 56.01 | 60.24 | 52.07 | 53.93 | 62.72 | **69.73** |

| Brightness | | | | | | | | | | |
|---|---|---|---|---|---|---|---|---|---|---|
| | KDAD | RD4AD | UnIAD | ViTAD | CFLOW-AD | patchcore | TTT++ | TTAC | GNL | **Ours** |
| audiojack | 36.81 | 85.30 | 79.88 | 75.51 | 65.84 | 78.25 | 35.97 | 38.26 | 85.90 | 81.40 |
| bottle_cap | 66.69 | 59.00 | 58.99 | 68.39 | 67.96 | 39.53 | 13.35 | 45.08 | 53.24 | 59.19 |
| button_battery | 36.41 | 80.90 | 72.60 | 65.89 | 67.41 | 86.40 | 43.74 | 57.48 | 79.87 | 81.40 |
| end_cap | 61.49 | 75.20 | 75.88 | 61.79 | 54.73 | 67.98 | 60.83 | 55.65 | 70.55 | 79.62 |
| eraser | 44.78 | 47.30 | 71.98 | 41.81 | 49.77 | 46.47 | 54.09 | 46.23 | 51.33 | 52.31 |
| fire_hood | 34.54 | 45.00 | 83.30 | 65.15 | 61.64 | 57.82 | 43.02 | 54.44 | 61.82 | 52.49 |
| mint | 49.79 | 61.70 | 56.69 | 46.79 | 49.68 | 65.08 | 44.42 | 66.31 | 51.05 | 56.66 |
| mounts | 31.38 | 72.70 | 75.42 | 56.62 | 59.27 | 68.59 | 59.06 | 59.45 | 57.02 | 73.79 |
| pcb | 38.97 | 78.10 | 81.46 | 65.04 | 46.06 | 63.62 | 40.16 | 66.87 | 71.90 | 75.08 |
| phone_battery | 28.61 | 77.30 | 80.39 | 79.15 | 50.19 | 83.22 | 41.82 | 86.30 | 87.18 | 81.35 |
| plastic_nut | 30.09 | 48.30 | 65.69 | 72.89 | 43.18 | 54.61 | 46.26 | 35.56 | 57.27 | 49.80 |
| plastic_plug | 66.79 | 65.40 | 72.07 | 68.09 | 49.34 | 46.33 | 28.91 | 63.08 | 73.59 | 81.73 |
| porcelain_doll | 69.06 | 38.50 | 37.40 | 26.75 | 57.35 | 37.03 | 53.41 | 37.11 | 53.51 | 35.04 |
| regulator | 49.63 | 43.30 | 51.35 | 45.96 | 40.24 | 58.43 | 49.25 | 45.40 | 55.49 | 46.94 |
| rolled_strip_base | 4.21 | 92.50 | 60.53 | 94.84 | 58.42 | 78.46 | 52.58 | 61.44 | 72.01 | 96.35 |
| sim_card_set | 26.78 | 82.60 | 80.20 | 51.79 | 60.04 | 70.80 | 38.19 | 18.50 | 89.51 | 88.46 |
| switch | 21.23 | 92.00 | 84.75 | 79.35 | 68.06 | 84.91 | 47.48 | 86.46 | 89.45 | 86.09 |
| tape | 47.35 | 62.50 | 81.31 | 49.91 | 65.89 | 68.92 | 33.72 | 60.25 | 65.75 | 75.02 |
| terminalblock | 32.67 | 41.80 | 46.88 | 55.97 | 48.41 | 53.42 | 34.05 | 39.29 | 46.25 | 63.94 |
| toothbrush | 39.53 | 41.90 | 64.97 | 60.20 | 62.50 | 55.31 | 39.06 | 62.67 | 56.06 | 74.36 |
| toy | 50.32 | 55.60 | 39.86 | 54.79 | 53.30 | 43.42 | 49.97 | 48.32 | 49.51 | 53.20 |
| toy_brick | 39.71 | 41.10 | 66.06 | 37.26 | 43.57 | 57.75 | 43.27 | 58.55 | 43.14 | 64.45 |
| transistor1 | 54.03 | 60.30 | 78.72 | 51.80 | 58.85 | 66.99 | 43.09 | 57.70 | 76.77 | 81.31 |
| u_block | 31.16 | 44.20 | 59.40 | 52.05 | 49.51 | 61.97 | 52.99 | 46.79 | 36.93 | 73.78 |
| usb | 15.52 | 87.90 | 90.63 | 73.84 | 60.43 | 83.26 | 59.23 | 38.44 | 88.92 | 93.11 |
| usb_adaptor | 24.30 | 67.20 | 74.38 | 74.19 | 53.51 | 65.20 | 51.89 | 68.80 | 64.42 | 65.49 |
| vcpill | 40.30 | 48.10 | 64.17 | 36.81 | 46.37 | 54.56 | 52.02 | 58.66 | 55.02 | 57.37 |
| wooden_beads | 27.93 | 52.00 | 63.42 | 59.41 | 45.00 | 36.51 | 33.17 | 35.72 | 44.92 | 47.13 |
| woodstick | 45.63 | 65.10 | 70.15 | 69.11 | 62.44 | 66.48 | 42.15 | 66.35 | 71.33 | 64.98 |
| zipper | 3.52 | 99.20 | 94.68 | 93.39 | 95.15 | 88.99 | 52.04 | 77.16 | 75.53 | 94.77 |
| mean | 38.31 | 63.73 | 69.44 | 61.15 | 56.47 | 63.01 | 44.64 | 54.74 | 64.51 | **69.55** |

| Defocus Blur | | | | | | | | | | |
|---|---|---|---|---|---|---|---|---|---|---|
| | KDAD | RD4AD | UnIAD | ViTAD | CFLOW-AD | patchcore | TTT++ | TTAC | GNL | **Ours** |
| audiojack | 18.94 | 89.00 | 82.32 | 85.93 | 46.51 | 90.03 | 28.63 | 59.79 | 86.36 | 89.28 |
| bottle_cap | 25.74 | 90.70 | 84.69 | 84.12 | 58.33 | 81.47 | 64.59 | 47.93 | 71.04 | 87.60 |
| button_battery | 58.03 | 80.00 | 69.68 | 67.50 | 61.96 | 75.24 | 41.04 | 74.67 | 76.40 | 76.73 |
| end_cap | 43.89 | 67.70 | 59.89 | 53.42 | 48.54 | 64.02 | 39.78 | 60.17 | 64.22 | 67.66 |
| eraser | 22.77 | 84.30 | 87.78 | 74.03 | 78.76 | 87.63 | 39.07 | 50.66 | 71.35 | 88.38 |
| fire_hood | 28.46 | 87.10 | 85.69 | 75.41 | 75.66 | 86.39 | 60.45 | 48.31 | 88.52 | 88.43 |
| mint | 50.81 | 51.90 | 54.25 | 56.43 | 52.47 | 61.01 | 51.66 | 58.66 | 55.34 | 61.45 |
| mounts | 17.12 | 88.40 | 84.46 | 83.43 | 71.54 | 83.68 | 55.00 | 70.66 | 67.10 | 92.08 |
| pcb | 38.35 | 60.00 | 72.79 | 75.07 | 42.28 | 77.21 | 53.33 | 68.09 | 80.51 | 88.53 |
| phone_battery | 21.45 | 47.80 | 79.78 | 84.63 | 65.25 | 71.81 | 30.01 | 51.67 | 88.01 | 83.87 |
| plastic_nut | 36.17 | 59.80 | 69.74 | 63.67 | 58.16 | 65.49 | 42.42 | 51.37 | 73.94 | 78.30 |
| plastic_plug | 19.11 | 86.90 | 79.39 | 84.40 | 76.09 | 84.12 | 42.82 | 81.14 | 82.04 | 86.90 |
| porcelain_doll | 24.59 | 72.40 | 67.29 | 68.61 | 71.04 | 67.48 | 34.97 | 69.34 | 73.75 | 80.66 |
| regulator | 33.97 | 73.50 | 50.29 | 45.29 | 50.42 | 58.21 | 55.31 | 52.45 | 83.71 | 70.82 |
| rolled_strip_base | 18.83 | 99.20 | 97.18 | 86.42 | 64.98 | 94.15 | 66.42 | 41.67 | 99.16 | 98.14 |
| sim_card_set | 14.32 | 90.10 | 90.50 | 65.56 | 74.03 | 96.98 | 68.57 | 70.37 | 94.20 | 96.46 |
| switch | 23.37 | 82.70 | 72.08 | 79.47 | 59.98 | 78.65 | 48.22 | 90.81 | 81.53 | 80.70 |
| tape | 46.45 | 94.70 | 97.51 | 86.27 | 80.22 | 96.34 | 53.92 | 42.62 | 93.26 | 98.18 |
| terminalblock | 46.65 | 89.10 | 76.19 | 87.29 | 50.05 | 67.10 | 56.24 | 74.58 | 88.07 | 90.04 |
| toothbrush | 26.63 | 83.70 | 90.19 | 81.01 | 47.88 | 78.22 | 60.01 | 53.34 | 82.23 | 83.22 |
| toy | 29.41 | 75.30 | 65.35 | 66.59 | 50.84 | 57.41 | 46.41 | 49.61 | 68.36 | 49.14 |
| toy_brick | 31.91 | 79.30 | 84.16 | 69.73 | 72.64 | 81.03 | 51.48 | 80.74 | 80.89 | 81.13 |
| transistor1 | 29.63 | 86.40 | 84.88 | 68.55 | 62.89 | 64.17 | 53.83 | 68.36 | 73.91 | 90.70 |
| u_block | 46.45 | 64.80 | 70.21 | 60.86 | 59.41 | 77.22 | 54.97 | 50.21 | 58.00 | 85.08 |
| usb | 21.80 | 79.60 | 83.68 | 80.60 | 58.96 | 67.16 | 45.41 | 39.80 | 84.89 | 82.51 |
| usb_adaptor | 32.76 | 71.50 | 73.35 | 62.16 | 62.72 | 69.32 | 53.80 | 71.35 | 61.19 | 70.63 |
| vcpill | 23.70 | 91.80 | 84.53 | 65.69 | 78.78 | 90.93 | 69.19 | 89.86 | 91.01 | 92.00 |
| wooden_beads | 19.29 | 79.80 | 85.40 | 73.93 | 65.49 | 80.91 | 51.49 | 42.77 | 86.16 | 82.85 |
| woodstick | 30.50 | 81.70 | 85.31 | 70.35 | 71.88 | 74.98 | 51.43 | 74.68 | 85.32 | 78.64 |
| zipper | 56.11 | 96.90 | 96.60 | 96.57 | 59.38 | 82.25 | 50.26 | 52.93 | 96.62 | 98.51 |
| mean | 31.24 | 79.54 | 78.84 | 73.43 | 62.57 | 77.02 | 50.69 | 61.29 | 79.57 | **83.29** |

|  | KDAD | RD4AD | UnIAD | ViTAD | CFLOW-AD | patchcore | TTT++ | TTAC | GNL | **Ours** |
|---|---|---|---|---|---|---|---|---|---|---|
| | | | | | Contrast | | | | | |
| audiojack | 29.56 | 61.4 | 25.78 | 68.89 | 53.23 | 56.31 | 52.82 | 43.08 | 81.06 | 71 |
| bottle_cap | 63.32 | 55.50 | 65.43 | 59.96 | 31.11 | 52.18 | 46.00 | 59.99 | 55.58 | 47.7 |
| button_battery | 67.56 | 49.70 | 46.67 | 57.88 | 50.32 | 50.91 | 55.77 | 49.41 | 58.62 | 56.42 |
| end_cap | 42.33 | 59.00 | 62.99 | 47.46 | 44.48 | 58.70 | 49.90 | 43.85 | 57.72 | 60.76 |
| eraser | 48.42 | 47.70 | 38.03 | 67.34 | 52.17 | 46.58 | 56.19 | 55.43 | 54.65 | 58.15 |
| fire_hood | 56.24 | 45.60 | 52.42 | 61.99 | 46.90 | 46.37 | 50.36 | 50.41 | 56.17 | 52.87 |
| mint | 46.95 | 40.30 | 49.72 | 50.81 | 44.62 | 48.16 | 49.67 | 55.72 | 59.65 | 52.57 |
| mounts | 51.75 | 62.80 | 39.96 | 50.18 | 53.76 | 50.49 | 48.84 | 46.07 | 55.71 | 61.19 |
| pcb | 43.38 | 83.10 | 46.40 | 55.26 | 42.35 | 54.31 | 59.17 | 60.93 | 45.32 | 64.8 |
| phone_battery | 24.66 | 77.10 | 35.88 | 56.08 | 57.19 | 30.47 | 54.49 | 70.54 | 67.11 | 51.09 |
| plastic_nut | 49.12 | 54.50 | 58.52 | 56.99 | 49.92 | 49.62 | 47.47 | 29.51 | 56.09 | 53.35 |
| plastic_plug | 72.42 | 55.10 | 71.50 | 71.88 | 63.06 | 59.55 | 69.98 | 77.24 | 77.1 | 74.48 |
| porcelain_doll | 40.93 | 54.70 | 57.34 | 48.58 | 46.67 | 57.77 | 40.92 | 60.40 | 69.51 | 53.31 |
| regulator | 39.61 | 55.40 | 50.81 | 55.39 | 50.27 | 50.97 | 54.51 | 50.76 | 56.59 | 58.7 |
| rolled_strip_base | 32.42 | 66.90 | 56.57 | 54.14 | 51.44 | 61.27 | 63.72 | 73.03 | 64.83 | 53.71 |
| sim_card_set | 62.88 | 46.00 | 62.67 | 51.30 | 79.17 | 39.38 | 45.27 | 87.36 | 64.52 | 67.04 |
| switch | 46.04 | 52.20 | 37.97 | 47.92 | 59.11 | 45.18 | 50.78 | 49.15 | 70.06 | 47.53 |
| tape | 29.75 | 76.40 | 82.01 | 58.92 | 58.60 | 51.45 | 36.27 | 51.70 | 83.44 | 87.46 |
| terminalblock | 63.91 | 46.20 | 49.97 | 49.93 | 66.72 | 64.56 | 52.65 | 41.13 | 66.25 | 66.31 |
| toothbrush | 69.30 | 47.70 | 51.29 | 57.42 | 60.75 | 40.18 | 42.36 | 69.07 | 66.39 | 62.32 |
| toy | 45.27 | 40.30 | 45.69 | 56.76 | 45.19 | 53.10 | 53.01 | 59.57 | 52.6 | 49.25 |
| toy_brick | 41.71 | 56.80 | 56.73 | 64.43 | 67.98 | 56.01 | 43.97 | 41.40 | 67.81 | 66.65 |
| transistor1 | 64.33 | 57.70 | 53.53 | 52.76 | 47.46 | 41.01 | 77.42 | 67.54 | 49.87 | 65.77 |
| u_block | 48.12 | 49.80 | 41.59 | 50.86 | 52.08 | 50.14 | 40.79 | 49.18 | 54.51 | 56.15 |
| usb | 42.87 | 47.20 | 50.69 | 52.42 | 48.41 | 49.96 | 43.38 | 45.61 | 70.58 | 60.18 |
| usb_adaptor | 47.54 | 56.60 | 65.62 | 47.01 | 47.59 | 43.78 | 47.41 | 43.16 | 48.31 | 64.99 |
| vcpill | 18.38 | 73.20 | 59.67 | 52.24 | 40.08 | 50.05 | 23.59 | 27.94 | 54.56 | 42.87 |
| wooden_beads | 56.03 | 58.90 | 59.24 | 57.70 | 43.49 | 49.98 | 51.79 | 51.18 | 60.47 | 57.34 |
| woodstick | 51.19 | 45.60 | 49.11 | 60.86 | 46.40 | 53.74 | 46.25 | 49.49 | 57.23 | 61.67 |
| zipper | 3.41 | 99.20 | 94.74 | 99.39 | 94.84 | 48.73 | 67.14 | 34.69 | 86.23 | 95.74 |
| mean | 46.65 | 57.42 | 53.95 | 57.43 | 53.18 | 50.36 | 50.73 | 53.15 | **62.28** | 60.71 |

Table 7: MVTec per class pixel AUROC(%)

| | Method | bottle | cable | capsule | carpet | grid | hazelnut | leather | metal_nut | pill | screw | tile | toothbrush | transistor | wood | zipper | mean |
|---|---|---|---|---|---|---|---|---|---|---|---|---|---|---|---|---|---|
| Gaussian Noise | Patch SVDD | 47.42 | 66.89 | 27.75 | 78.95 | 78.22 | 82.63 | 81.76 | 41.39 | 52.78 | 21.52 | 48.49 | 19.96 | 44.19 | 80.54 | 57.85 | 55.36 |
| | RD4AD | 54.40 | 97.12 | 93.68 | 98.28 | 97.45 | 96.00 | 99.08 | 63.01 | 73.49 | 95.80 | 91.45 | 97.25 | 82.54 | 88.20 | 75.5 | 86.88 |
| | CFLOW-AD | 67.84 | 77.17 | 73.86 | 73.11 | 57.56 | 63.65 | 72.42 | 72.07 | 69.03 | 84.21 | 80.63 | 51.71 | 90.57 | 74.17 | 47.70 | 70.38 |
| | patchcore | 84.97 | 92.74 | 93.67 | 95.61 | 67.76 | 96.10 | 97.01 | 84.51 | 76.2 | 88.94 | 83.48 | 94.35 | 84.14 | 80.10 | 85.44 | 87.00 |
| | TTT++ | 78.44 | 89.74 | 88.00 | 95.41 | 65.47 | 81.69 | 95.66 | 75.80 | 55.23 | 79.79 | 84.50 | 94.63 | 80.61 | 81.53 | 58.18 | 80.31 |
| | TTAC | 74.87 | 49.28 | 43.88 | 51.21 | 62.27 | 58.52 | 50.46 | 66.97 | 48.93 | 43.78 | 53.86 | 75.11 | 66.29 | 47.79 | 58.28 | 56.77 |
| | **Ours** | 95.98 | 96.47 | 94.98 | 97.68 | 94.2 | 98.36 | 98.44 | 93.77 | 87.07 | 95.73 | 89.25 | 97.98 | 90.41 | 88.19 | 93.46 | **94.13** |
| Brightness | Patch SVDD | 73.07 | 54.15 | 36.05 | 53.62 | 46.68 | 84.51 | 55.91 | 75.84 | 50.08 | 80.69 | 57.77 | 79.17 | 61.90 | 49.86 | 79.03 | 62.56 |
| | RD4AD | 98.79 | 95.92 | 79.74 | 99.13 | 98.31 | 100.00 | 100.00 | 92.31 | 89.70 | 61.26 | 97.74 | 68.95 | 87.78 | 99.75 | 93.13 | 90.83 |
| | CFLOW-AD | 91.94 | 93.06 | 90.11 | 71.76 | 45.90 | 75.57 | 81.91 | 67.87 | 75.30 | 22.99 | 89.99 | 74.72 | 85.61 | 88.12 | 83.17 | 75.87 |
| | patchcore | 97.20 | 95.54 | 95.73 | 97.57 | 90.33 | 97.36 | 98.10 | 95.98 | 85.38 | 51.14 | 92.29 | 97.16 | 90.60 | 90.12 | 96.41 | 91.39 |
| | TTT++ | 96.45 | 92.38 | 73.95 | 96.02 | 65.51 | 70.36 | 94.70 | 93.59 | 59.08 | 22.76 | 90.78 | 97.52 | 84.01 | 80.13 | 70.79 | 79.20 |
| | TTAC | 98.22 | 96.52 | 45.60 | 96.92 | 91.94 | 98.04 | 97.78 | 75.83 | 66.99 | 21.14 | 74.85 | 97.52 | 88.66 | 87.76 | 96.48 | 82.28 |
| | **Ours** | 97.69 | 97.26 | 96.12 | 98.77 | 97.10 | 98.10 | 99.02 | 97.15 | 89.77 | 97.09 | 93.83 | 98.33 | 94.99 | 92.96 | 98.40 | **96.44** |
| Defocus Blur | Patch SVDD | 80.79 | 33.55 | 93.92 | 35.12 | 67.48 | 75.10 | 46.73 | 69.72 | 49.95 | 63.20 | 59.98 | 85.10 | 74.82 | 82.21 | 57.70 | 65.02 |
| | RD4AD | 98.30 | 97.22 | 94.77 | 98.30 | 97.12 | 98.84 | 99.47 | 97.53 | 97.06 | 98.50 | 93.13 | 98.29 | 89.87 | 93.65 | 95.82 | 96.52 |
| | CFLOW-AD | 94.01 | 93.47 | 60.77 | 74.39 | 62.26 | 98.08 | 80.92 | 92.90 | 89.12 | 40.00 | 80.44 | 83.58 | 95.26 | 70.80 | 71.85 | 79.19 |
| | patchcore | 97.06 | 96.68 | 97.54 | 95.90 | 75.49 | 98.55 | 98.82 | 94.48 | 93.17 | 93.79 | 89.85 | 96.80 | 93.08 | 89.73 | 89.21 | 93.34 |
| | TTT++ | 80.59 | 91.83 | 80.86 | 94.47 | 62.72 | 85.85 | 94.73 | 75.88 | 47.08 | 69.94 | 84.54 | 94.63 | 73.34 | 81.31 | 70.41 | 79.21 |
| | TTAC | 40.94 | 76.34 | 27.30 | 65.28 | 51.07 | 27.66 | 24.59 | 34.94 | 19.03 | 8.84 | 60.01 | 23.33 | 62.56 | 39.07 | 15.55 | 38.43 |
| | **Ours** | 97.71 | 97.36 | 97.98 | 97.67 | 94.10 | 98.64 | 99.06 | 95.97 | 95.05 | 97.4 | 93.15 | 98.34 | 95.58 | 92.48 | 97.44 | **96.53** |

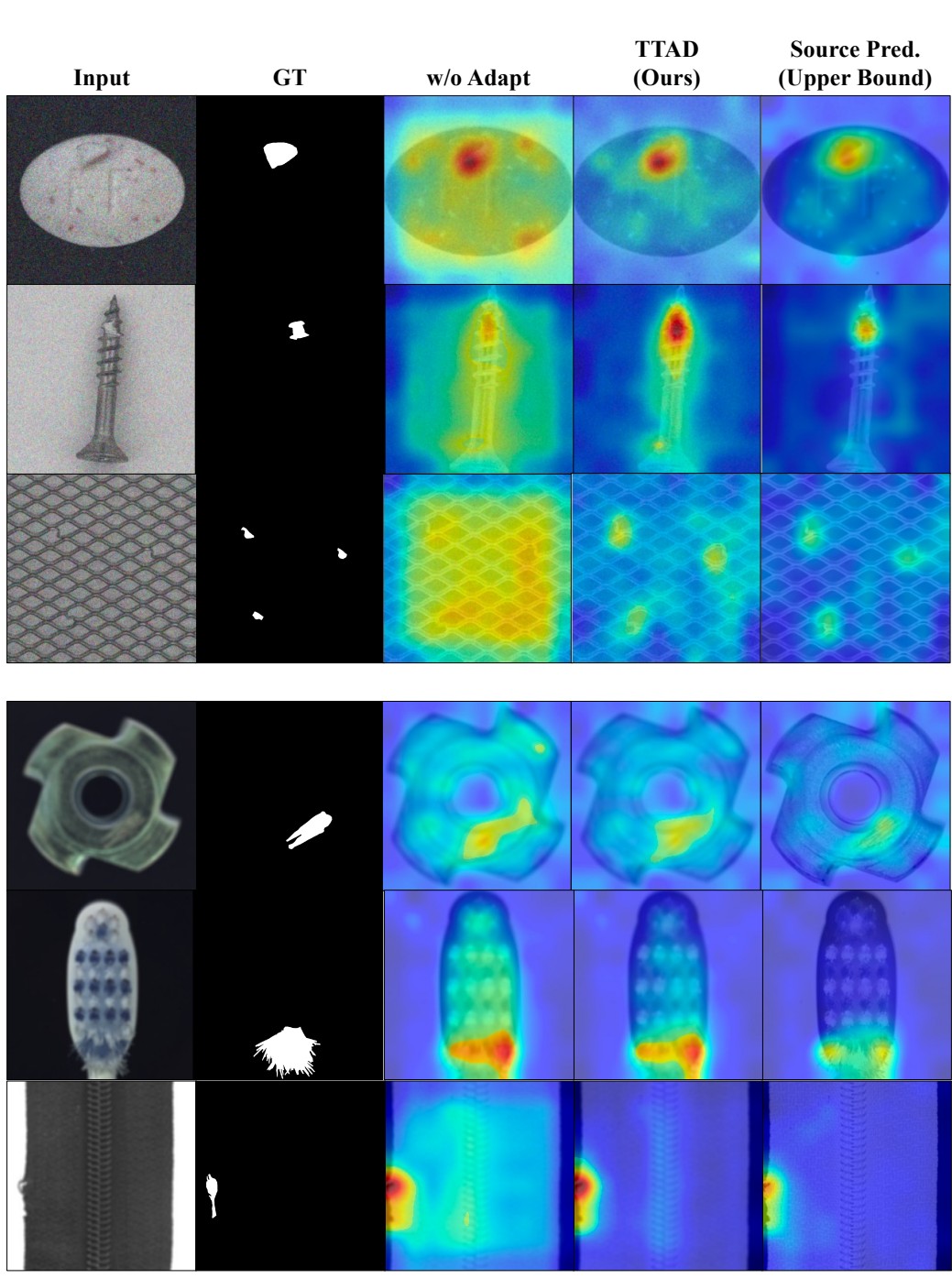

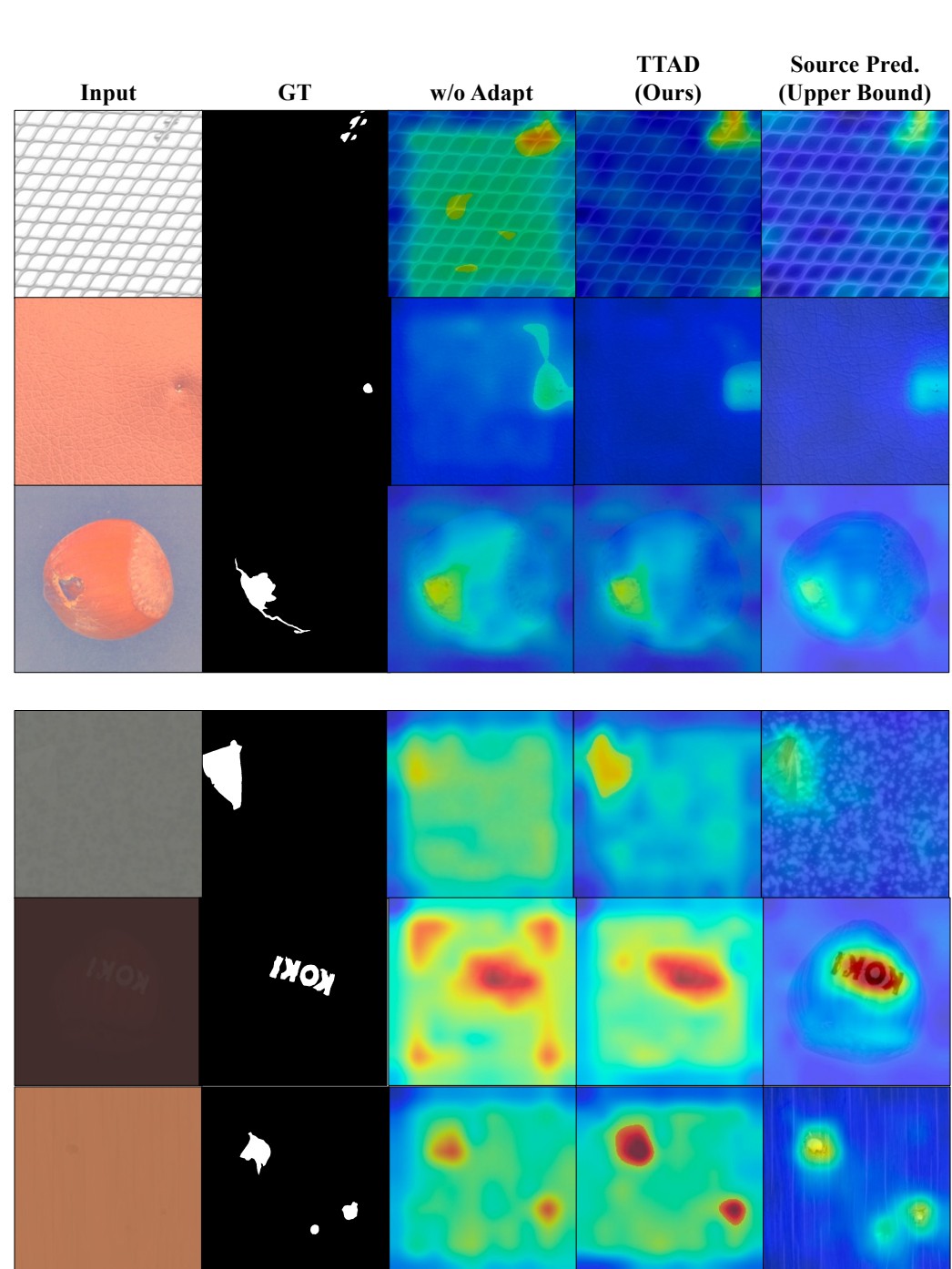

Figure 8: More qualitative segmentation results from MVTec dataset.

