# OpenReview forum: "Test-Time Training for Out-of-Distribution Industrial Anomaly Detection via Robust Distribution Alignment"
_ICLR.cc/2025/Conference — ICLR 2025 Conference Withdrawn Submission_

### Official Review · Reviewer_Eqg3 · 2024-10-20

**Soundness:** 2
**Presentation:** 1
**Contribution:** 1
**Rating:** 1
**Confidence:** 4

**Summary:**

The authors aim to address Industrial Anomaly Detection under distributional shifts by proposing a memory bank-based anomaly detection method. This approach seeks to reduce reliance on prior knowledge of target distributions and eliminate the need for access to training data during inference.

**Strengths:**

The authors effectively apply advanced techniques to Industrial Anomaly Detection, enhancing the model's ability to handle distributional shifts.

**Weaknesses:**

Lack of Importance and Novelty: The problem that the paper aims to solve does not seem particularly novel or significant. The paper attempts to address two issues faced by GNL. First, requiring prior knowledge of the target data distribution during training may not be practical. GNL's performance can degrade significantly when the distribution shift at test time is different from the augmentations used during training. Second, accessing normal training samples during inference may be restricted due to privacy concerns or data storage limitations. However, both of these problems have already been extensively researched and addressed by prior works. For instance, the first problem has been well-studied, and methods such as Invariant Risk Minimization can be used to tackle it.

Insufficient Explanation of the Proposed Method's Effectiveness: The paper does not clearly articulate why the proposed method is effective in solving the identified issues. On line 76, the authors state that a memory bank is introduced to solve these problems. However, they do not adequately explain why the memory bank specifically helps address the challenge of distribution shifts during test time.

Poor Writing Quality and Logical Gaps: There are numerous flaws in the logical flow of the paper, making it difficult to understand. Here are some specific examples:

a. On line 82, the authors mention referring to Fig. 2 to observe performance decline, but this is not clearly visible from the figure.

b. On line 84, test-time adaptation is introduced without sufficient motivation, making the reasoning difficult to follow.

c. On line 92, the authors argue that mixture Gaussian KL-Divergence lacks a closed-form solution and thus cannot be used. However, mixture Gaussian models are widely applied, and with appropriate adjustments, this issue could be addressed.

d. On line 94, it is unclear why test-time training is considered an optimal transport problem.

As illustrated above, the paper contains many subjective viewpoints without concrete explanations. The issues listed here represent only part of the logical gaps in the paper. I believe the authors need to further refine and revise their work.

Unclear and Potentially Erroneous Mathematical Formulation: The mathematical formulations in the paper are not clearly presented, and some even appear to contain errors, further hindering comprehension.

a. On line 172, the authors refer to a backbone network as $z_i$, but $z_i$ clearly cannot represent the backbone network.

b. On line 173, $C(\cdot, K)$ is introduced, but neither $C$ nor $K$ is defined.

c. In Equation 1, the authors denote $M \in D_s$, but there is no apparent set relationship between $M$ and $D_s$. Similarly, there is no clear set relationship between $z_j$ and $D_s$.

Such errors are not uncommon in the paper, making it increasingly difficult to follow.

An Unnatural Combination of Techniques: The proposed approach appears to be a somewhat forced combination of two existing techniques aimed at solving the Industrial Anomaly Detection problem, lacking originality.

**Questions:**

N/a

---

### Official Review · Reviewer_32W1 · 2024-11-01

**Soundness:** 4
**Presentation:** 3
**Contribution:** 4
**Rating:** 8
**Confidence:** 2

**Summary:**

This paper introduces a novel test-time training (TTT) approach by leveraging robust distribution alignment (DA) using optimal transport. The approach primarily employs a memory bank and optimal transport-based DA to align distributions, using a modified Sinkhorn distance to mitigate the effects of OOD patches. Extensive experiments on multiple industrial anomaly detection datasets under various distribution shift scenarios demonstrate the robustness and effectiveness of TTAD, with substantial improvements over existing methods.

**Strengths:**

- The paper addresses a critical issue in anomaly detection resolving distribution shifts during inference.
- The use of Sinkhorn distance is fairly technical and is suitable.
- Augmentation in the target domain at test time is clever.
- The paper includes extensive benchmarking against state-of-the-art anomaly detection and TTT methods on a variety of datasets.

**Weaknesses:**

- While the paper addresses computational efficiency via Sinkhorn regularization, the approach’s scalability as memory bank size increases could be challenging.
- Due to the importance of replication of results for development of research community it is important that any implementation became publicly available. It is recommended that authors use "anonymous for open science" (anonymous GitHub) for their implementation.

**Questions:**

1- I would like a discussion on the trade-offs between memory bank size, computational cost, and performance. Addition of complexity analysis would be appropriate.

---

### Official Review · Reviewer_pDKx · 2024-11-02

**Soundness:** 2
**Presentation:** 3
**Contribution:** 2
**Rating:** 3
**Confidence:** 5

**Summary:**

In general, this work introduces an interesting test-time training (TTT) pipeline for industrial anomaly detection (IAD) task, which in my view should be valuable and crucial for  few-shot IAD. However, in this work, the authors introduces TTT in traditional IAD, the motivation might be unreasonable and unclear. To realize such TTT pipeline, the authors propose a rectified Sinkhorn algorithm to make distribution alignment between normal memory and online query images. Experiments on MVTec and RealIAD datasets verify the effectiveness of the proposed method.

**Strengths:**

1. As far as I know, this work is the first one to introduce TTT in traditional IAD task, it is a kind of interesting.
2. Rectified Sinkhorn algorithm with discretizing the original one is a little interesting.

**Weaknesses:**

Despite the above strengths, the weaknesses are significant.
1. The motivation of introducing TTT in traditional IAD task might be unreasonable and incorrect. In the practical industrial production line, the imaging acquisition equipment is always high quality and thus there does not exist noising input as the authors discussed and verified in their main experimental results, such as Table 1 and Table 3. Therefore, a more reasonable way to employ TTT should be in few-shot IAD, where the normal memory is easily overfitted and need to be refined. The most recently corresponding works are depicted in [1].
2. There lacks theoretical analysis on convergence of the proposed rectified Sinkhorn algorithm.
3. There lacks comparisons on real-time efficiency between the proposed method and other competitors, which should be crucial for practical use.

[1] FOCT: Few-shot industrial anomaly detection with foreground-aware online conditional transport.

**Questions:**

Please see the weaknesses above.

---

### Official Review · Reviewer_CJF4 · 2024-11-04

**Soundness:** 2
**Presentation:** 1
**Contribution:** 2
**Rating:** 3
**Confidence:** 3

**Summary:**

Their paper proposes a new industrial anomaly detection method. While existing systems often fail when conditions change, and previous work (GNL) attempted to solve this during training, their memory bank approach adapts to changes in real-time without needing training data. Their tests show this method handles shifting conditions more effectively than current solutions.

**Strengths:**

The authors performed comprehensive experiments.

**Weaknesses:**

- This appears to be primarily an engineering challenge rather than an academic research problem.
- The methodology would benefit from stronger theoretical foundations, as the current approach seems ad-hoc.
- The manuscript requires editorial attention, particularly regarding citation formatting in the main text and proper quotation marks when converting from other document formats.

**Questions:**

See weaknesses.

---

### Note · Authors · 2024-11-15

I have read and agree with the venue's withdrawal policy on behalf of myself and my co-authors.